# Enhancing the Clinical Utility of Radiomics: Addressing the Challenges of Repeatability and Reproducibility in CT and MRI

**DOI:** 10.3390/diagnostics14161835

**Published:** 2024-08-22

**Authors:** Xinzhi Teng, Yongqiang Wang, Alexander James Nicol, Jerry Chi Fung Ching, Edwin Ka Yiu Wong, Kenneth Tsz Chun Lam, Jiang Zhang, Shara Wee-Yee Lee, Jing Cai

**Affiliations:** 1Department of Health Technology and Informatics, The Hong Kong Polytechnic University, 11 Yuk Choi Rd, Hung Hom, Hong Kong SAR, China; xinzhi.teng@connect.polyu.hk (X.T.); wyq331@mail.ustc.edu.cn (Y.W.); alexander.nicol@connect.polyu.hk (A.J.N.); jerrycf.ching@connect.polyu.hk (J.C.F.C.); 24004171g@connect.polyu.hk (E.K.Y.W.); b101107132@tmu.edu.tw (K.T.C.L.); jiang.j.zhang@polyu.edu.hk (J.Z.); 2Hong Kong Polytechnic University Shenzhen Research Institute, Shenzhen 518057, China

**Keywords:** radiomics, repeatability and reproducibility

## Abstract

Radiomics, which integrates the comprehensive characterization of imaging phenotypes with machine learning algorithms, is increasingly recognized for its potential in the diagnosis and prognosis of oncological conditions. However, the repeatability and reproducibility of radiomic features are critical challenges that hinder their widespread clinical adoption. This review aims to address the paucity of discussion regarding the factors that influence the reproducibility and repeatability of radiomic features and their subsequent impact on the application of radiomic models. We provide a synthesis of the literature on the repeatability and reproducibility of CT/MR-based radiomic features, examining sources of variation, the number of reproducible features, and the availability of individual feature repeatability indices. We differentiate sources of variation into random effects, which are challenging to control but can be quantified through simulation methods such as perturbation, and biases, which arise from scanner variability and inter-reader differences and can significantly affect the generalizability of radiomic model performance in diverse settings. Four suggestions for repeatability and reproducibility studies are suggested: (1) detailed reporting of variation sources, (2) transparent disclosure of calculation parameters, (3) careful selection of suitable reliability indices, and (4) comprehensive reporting of reliability metrics. This review underscores the importance of random effects in feature selection and harmonizing biases between development and clinical application settings to facilitate the successful translation of radiomic models from research to clinical practice.

## 1. Introduction

In the era of precision medicine, translational research that aims to solve specific clinical questions with technical developments has gained increasing popularity due to the availability of structured medical data and rapid advancements in data mining techniques. Imaging is one of the most frequently analyzed data modalities due to the wide availability of imaging data and their rich anatomical, textural, and functional information. Image-derived biomarkers have been used in routine clinical practice, such as the TNM stage determined from multiple imaging modalities and the bone scan index calculated from SPECT [1]. Meanwhile, new imaging biomarkers have been actively investigated to fully explore the potential of imaging data in personalized clinical decision making. One of the most popular quantitative imaging biomarker development techniques is radiomics, in which a comprehensive set of features are extracted from medical images and are correlated with the underlying pathophysiology [2]. The general workflow of radiomics has been shown in Figure 1. It has been recognized as a potentially effective tool for diagnosis, survival prognosis, and toxicity prediction by combining selected features into single predictive signatures [3,4,5,6,7].

Despite the large number of radiomic signatures proposed in previous studies, very few have been externally validated in a prospective setting [8], posing a great challenge to reliable clinical applications. A repeatable and reproducible radiomic signature requires complete transparency in signature composition and consistent measurements in the same or similar conditions [9]. The latter, consistent measurement, relies heavily on the repeatability and reproducibility of individual radiomic features, which can be affected by inconsistencies during all the steps of radiomic feature acquisition, including image acquisition, structure delineation, image preprocessing, and feature extraction. The Imaging Biomarker Standardization Initiative (IBSI) attempted to achieve consensus on the settings and procedures in image preprocessing and feature extraction through international collaboration and provided guidelines on quality assurance and feature reporting [10]. On the other hand, absolute agreements in image acquisition and structure delineation are infeasible due to heterogeneous machines and imaging acquisition protocols, randomness in machine status and patient setups, as well as bias and error in manual structure delineations. Most reviews have approached this issue by summarizing and analyzing the reliability of radiomic features with respect to each aspect and have provided suggestions to mitigate the repeatability issues in each case. More clinical radiomic studies have assessed radiomic feature repeatability and reproducibility under these types of uncertainties using experimental techniques, including test–retest imaging and multiple delineations [11,12,13,14], and perturbation methods [15,16,17].

It has been recognized that each step in the radiomic workflow impacts on radiomic feature reliability, and several review articles have comprehensively investigated the source of variations affecting radiomic feature repeatability in radiomic workflows. Zhao [18] discussed the sources of variations in the radiomic workflow, categorizing them into controllable and uncontrollable factors to provide a deeper perspective on the reliability of radiomics and also provide potential solutions for each step. Yet, despite plenty of discussions on the radiomic workflow, most reviews conducted a radiomic workflow-based evaluation of sources of variations, and there was a lack of discussion on the nature of the sources of variations. Furthermore, most of the reviews summarized the sources of variations, and there was a lack of discussion of the results. Additionally, focusing on individual feature repeatability would be more practical for developing reliable radiomic signatures. For example, the assessment of radiomic feature repeatability via test–retest and perturbation methods has been proven to be valuable in safeguarding the reliability of radiomic signatures by improving their internal and external generalizability when developed using only repeatable radiomic features.

Therefore, this review aimed to (1) summarize the sources of variation in the CT/MR-based radiomic feature repeatability literature in terms of random effects and bias and discuss their implications for different applications and (2) summarize the number/proportion of highly repeatable and reproducible radiomic features under randomness and bias in radiomic workflows reported in previous studies. Specifically for the second arm, we focused on the comparison of highly repeatable and reproducible features under different sources of variations, including scanners, image acquisition protocols, test–retest imaging, intra-observer contouring, and inter-observer contouring, categorized by different imaging modalities. This review could provide a more holistic picture of radiomic feature susceptibility to different bias and random factors during applications in real clinical scenarios and facilitate the construction of reliable radiomic signatures.

## 2. Materials and Methods

### 2.1. Eligibility Criteria

Peer-reviewed full-text articles written in the English language and published between 1 May 2017 and 1 December 2023 were eligible for inclusion in this review. Three electronic databases (PubMed, EMbase, and Web of Science) were used to search for records. Publications included in our review met all of the following inclusion criteria: (1) peer-reviewed English full-text reports; (2) included radiomic features extracted from CT images or MR images; (3) indicated compliance with IBSI during feature extraction; and (4) focused on the repeatability and reproducibility of radiomic features resulting from variations during image acquisition and segmentation.

### 2.2. Research Strategy

To search for articles, we used the following search string: (“Radiomics” OR “Texture”) AND (“Reproducibility” OR “Repeatability” OR “Robustness” OR “Stability”) AND (“CT” OR “CT Scan” OR “Computed Tomography” OR “MRI” OR “Magnetic resonant image”). We also screened the Cochrane Database of Systematic Reviews for any previous reviews addressing the robustness of CT-based radiomic features. For all the articles obtained where we used the full text for data extraction, we screened the bibliographic references within them for additional potentially eligible studies. The researchers downloaded these electronic full-text articles using university library subscriptions. Two experienced researchers independently reviewed the eligibility of the studies based on the previously mentioned eligibility criteria.

### 2.3. Data Extraction

Data were extracted to a spreadsheet with a drop-down list for each item, defined by the first author, imaging modalities, sources of variations, criteria for highly repeatable/reproducible features, imaging sites, number/portion of highly repeatable/reproducible features, and availability of repeatability/reproducibility metric values for individual features.

## 3. Results

### 3.1. Overall Results

Overall, 38 publications, including 24 publications on CT scans and 16 publications on MR scans (2 publications involved both CT and MR scans), were included in the analysis. The inclusion flowchart was shown in Figure 2.

The sources of variations were categorized into two main categories, random effects and bias. For sources of variations, random effects were summarized according to (1) intra-scanner test–retest (CT: *n* = 7, MR: *n* = 6), (2) intra-observer variability on segmentations (CT: *n* = 2, MR: *n* = 1), (3) perturbations (CT: *n* = 1, MR: *n* = 0), and (4) auto-generated segmentations (CT: *n* = 1, MR: *n* = 0). Bias was summarized according to (1) acquisition parameters (CT: *n* = 9, MR: *n* = 2), (2) inter-scanner variability (CT: *n* = 4, MR: *n* = 5), (3) inter-observer variability on segmentations (CT: *n* = 6, MR: *n* = 5), (4) preprocessing parameters (CT: *n* = 3, MR: *n* = 3), (5) contrast agent-related bias (CT: *n* = 2, MR: *n* = 0), and (6) spatial variability (CT: *n* = 0, MR: *n* = 1).

For the reliability index quantifying feature repeatability against random effects, concordance coefficients of correlation (CCCs) (*n* = 5), intra-class coefficients of correlation (ICCs) (*n* = 11), coefficients of variance (CVs) (*n* = 2), dynamic ranges (DRs) (*n* = 1), and Bland–Altman analysis (*n* = 1) were used. Several publications [19,20,21] applied a combined index to identify highly repeatable features. For the reliability index quantifying feature reproducibility against bias, CCCs (*n* = 13), ICCs (*n* = 20), CVs (*n* = 8), quartile coefficients of dispersion (QCDs) (*n* = 1), DRs (*n* = 2), Kendall’s concordance coefficients (*n* = 2), r^2^ (*n* = 1), and average symmetric mean absolute percentage errors (*n* = 2) were used. Similarly, several publications [19,20,22,23,24,25,26] applied a combined reliability index to identify highly repeatable features. Furthermore, the thresholds for identifying repeatable/reproducible features also exhibited relatively large variations across different studies.

The imaging sites covered in the included studies were phantom (*n* = *5*), liver (*n* = 5), lung (*n* = 6), skin (*n* = 1), heart (*n* = 1), pancreas (*n* = 1), and kidney (*n* = 2) sites for CT scans. The imaging sites covered in the included studies were phantom (*n* = 1), brain (*n* = 4), lung (*n* = 1), liver (*n* = 3), heart (*n* = 1), breast (*n* = 1), prostate (*n* = 2), and cervix (*n* = 1) sites for MRI scans.

The proportion of features found to be highly repeatable against random effects ranged from 15.1% to 93.1% across the literature investigating feature repeatability on CT scans and from 0.50% to 91.6% across the literature investigating feature repeatability on MR scans. The proportion of features that were highly reproducible against bias ranged from 0% to 100% across the literature on CT scans and from 2.5% to 96.7% across that on MR scans. Furthermore, a clear trend was observed, namely, that more features are susceptible to inter-scanner/observer variability than intra-scanner/observer variability. 

Lastly, 26 out of 38 included publications had repeatability/reproducibility indices available for individual radiomic features.

### 3.2. Random Effects Affecting CT-Based Radiomic Features

Ten publications were included in the summary of the role of random effects on the repeatability of CT-based radiomic features, as shown in Table 1.

### 3.3. Bias Affecting CT-Based Radiomic Features

Twenty-one publications were included in the summary of the effect of bias on the repeatability of CT-based radiomic features, as shown in Table 2.

### 3.4. Random Effects Affecting MR-Based Radiomic Features

Eight publications were included in the summary of the role of random effects in the repeatability of MRI-based radiomic features, as shown in Table 3.

### 3.5. Bias Affecting MR-Based Radiomic Features

Thirteen publications were included in the summary of investigations on MRI-based radiomic feature repeatability affected by random effects, as shown in Table 4.

## 4. Discussion

### 4.1. Significance of Repeatability and Reproducibility in Radiomic Studies

Radiomics has emerged as a pivotal technique to augment the value of medical imaging through high-throughput characterization of medical images. The explicit mathematical definitions of each radiomic feature enhance the interpretability of radiomic signatures, offering a more transparent alternative to the “black box” deep learning models. The key advantage of radiomics over deep learning-based methods is the standardization of the image preprocessing and feature definition standardization due to the effort by Zwanenburg et al. An increasing number of publications have explored the diagnostic and prognostic value of radiomic approaches for various diseases in recent years. However, despite the surge in publications, concerns about reproducibility and repeatability have been prevalent since the inception of the field. It is believed that the usage of highly repeatable and reproducible radiomic features should be the first and foremost criterion to safeguard downstream model reliability [16]. Previous evidence has also suggested the positive impact of repeatable radiomic features in improving both the internal and external generalizability of radiomic models [6,17,52]. Guidelines such as the EvaluAtion of Radiomics research (CLEAR) [53], the Radiomics Quality Score (RQS) [54], and the Joint EANM/SNMMI guideline on radiomics [55] have underscored the importance of evaluating the reproducibility and repeatability of radiomic features and models. Understanding the source of radiomic feature variability is fundamental to harnessing the potential of radiomics in precision medicine by ensuring its reliability and determining its scope of application.

### 4.2. Randomness: A Fundamental Source of Variation in Radiomic Studies

Randomness, inherent and unpredictable variability which cannot be controlled in image acquisition and segmentation, is a primary concern when addressing repeatability issues in radiomic studies. The influence of randomness on radiomic features has been the subject of extensive research, particularly through the use of repeated scans with identical scanners at brief intervals. This randomness can arise from factors such as patient positioning and scanner noise, which may induce fluctuations in image intensity and, as a result, affect the consistency of radiomic features. For instance, Muenzfeld et al. [29] investigated the repeatability of CT-based radiomic features by performing multiple scans on a medical phantom with the same scanner, applying a CCC threshold of 0.85 to define repeatability. Their study revealed that a mere 22% (19 out of 86) of the features from original images met this repeatability criterion. Similarly, other research has explored the effects of randomness on radiomic features, particularly with intra-observer segmentations, where a single observer is responsible for multiple delineations on the same subject. Here, the randomness is attributed to the variability in segmentation boundaries, which affects the region of interest for feature extraction and, consequently, the radiomic features themselves. Specifically, Duan et al. [31] examined the impact of intra-observer variability by setting a more permissive threshold for high repeatability, with an ICC greater than 0.75. Their results indicated that 78.5% (84 out of 107) of CECT-based radiomic features and 72.0% (77 out of 107) of CT-based radiomic features were repeatable. It is important to recognize that intra-observer variability in ROI delineation may differ depending on the imaging modality, the anatomical site, and the observer’s experience. These studies highlight the vulnerability of radiomic features to randomness. The implications are clear: employing non-repeatable features to construct a radiomic signature can render the signature susceptible to randomness, potentially leading to a significant margin of error in its prognostic or diagnostic utility. Therefore, ensuring the repeatability of radiomic features is essential for the development of robust and reliable radiomic signatures.

### 4.3. Bias: Inter-Observer and Inter-Scanner Variations—A Significant Hurdle to Generalizable Radiomic Signatures

Variations in the measurement settings of radiomic features can significantly impact their consistency. These variations can stem from changes in acquisition protocols or cross scanners, or from segmentations being performed by different observers. It is crucial to distinguish variations that are not random but are instead attributable to the inherent biases associated with different scanners or observer practices. Inherent bias refers to systematic differences that are difficult to replicate during applications. For example, a radiomic signature developed using data from a specific scanner and radiologist is likely to underperform in a new institution with a different scanner and radiologist.

Radiomic signatures often demonstrate optimal validation performance within the clinical settings in which they were developed. However, this performance can deteriorate when they are applied in different scenarios, such as when using alternative scanner brands or imaging protocols, or when segmentation is conducted by different observers. The further the application deviates from the original clinical setting, the more pronounced the decline in generalizability becomes, as evidenced by diminished discrimination performance.

The impact of inter-observer and inter-scanner variability on the reproducibility of radiomic features has been the focus of several studies. It has been consistently observed that radiomic features are more vulnerable to variations introduced by different observers or scanners than to those arising from the same observer or scanner. For example, Fiset et al. [13] examined the repeatability and reproducibility of MR-based radiomic features in the context of inter-scanner and intra-scanner rescans. Their findings indicated that while 52.1% of radiomic features remained reproducible in the face of intra-scanner variability, a mere 14.1% maintained reproducibility when confronted with inter-scanner variability. A similar pattern emerged when comparing intra-observer and inter-observer variability.

Understanding the distinction between random effects and bias is imperative for the reporting of results. The metrics used to measure repeatability and reproducibility should differ. High repeatability against random effects is typically defined by the absolute agreement between repeated measures, whereas consistency measures are more appropriate for defining reproducibility in the presence of bias. Koo et al. [56] provided a practical guideline for applying the intra-class correlation coefficient (ICC) in assessing test–retest reliability, inter-rater reliability, and intra-rater reliability. This guideline facilitates the appropriate use of ICCs to account for both random effects and bias, thereby enhancing the reliability of radiomic feature measurements.

### 4.4. Efforts to Mitigate Randomness for Repeatable Radiomic Signatures

The primary goal in mitigating the effects of randomness on radiomic features is to develop a robust radiomic signature that can consistently deliver the same results, irrespective of random fluctuations. To achieve this, two key strategies should be employed. First, the extraction process should be refined to maximize the number of reproducible features, which involves optimizing image preprocessing parameters such as interpolations, rounding intensities, and outlier filters. Second, the construction of a radiomic signature should be based on features that have demonstrated repeatability and resilience to random variations.

Dewi et al. [47] conducted a study to assess the repeatability of features under various preprocessing conditions on T2-weighted MR images. They pinpointed a specific set of preprocessing parameters, namely, a fixed bin count of 64, the absence of signal intensity normalization, and the exclusion of outliers, which resulted in the highest number of repeatable features. However, this raises a critical question: Is the pure quantity of repeatable features the most reliable indicator of optimal preprocessing settings, or should the sensitivity of radiomic features to preprocessing also be taken into account?

Moreover, the construction of radiomic signatures should prioritize the inclusion of repeatable features. Teng et al. [17] evaluated multiple radiomic signatures by systematically excluding features with low repeatability, applying ICC thresholds of 0, 0.5, 0.75, and 0.95. Their findings indicated that increasing the threshold for feature repeatability not only enhanced the repeatability of the radiomic signatures but also maintained their discriminative capability. Similarly, Zhang et al. [52] showed that the exclusion of features with low repeatability from the signature construction process improved the inter-institutional generalizability of the radiomic model.

However, assessing the repeatability of radiomic features in the face of randomness presents a significant challenge, as gold-standard test–retest scans are often not readily available. The repeatability determined from a limited set of test–retest scans may not be universally applicable to other datasets. To address this data scarcity, Zwanenburg et al. [15] introduced a simulation-based approach as an alternative to actual test–retest scans. This method generates pseudo-test–retest scans by applying transformations such as rotation, translation, and noise addition to the original images, along with contour randomizations at the edges of segmentations. The robust features identified through this simulation technique align with those found to be repeatable in actual test–retest scenarios. Building on this, Zhang et al. [57] further validated that simulation methods for identifying repeatable features can lead to the development of generalizable radiomic signatures comparable to those derived from test–retest scans. This suggested the potential of simulation-based methods as a viable solution for overcoming the limitations posed by the scarcity of test–retest data.

### 4.5. Efforts to Address Bias for Generalizable Radiomic Signatures

To ensure the generalizability of a radiomic model, it is critical that it retains its discriminative ability across diverse clinical settings. This entails maintaining performance despite potential variations, such as differences in scanner brands, raw data acquisition protocols, and the methodologies employed by radiologists or physicians in delineating regions of interest. While these factors are not inherently random, they are often difficult to control when applying radiomic signatures in practice. Thus, the approach to mitigating these issues parallels that of addressing randomness, namely, identifying and utilizing radiomic features that are robust to the variations likely to be encountered in real-world application scenarios. Hoebel et al. [58] reported that normalization and intensity quantization can affect the level of repeatability of radiomic features. Moradmand et al. [26] evaluated various combinations of preprocessing steps for multi-parametric MR images and found that a sequence of bias field correction followed by noise filtering produced the most reproducible radiomic features.

Three primary sources of variation must be considered: the scanner used, the image acquisition protocol, and inter-observer variability in contouring. Of these, inter-observer variability can be mitigated by involving multiple observers in the segmentation process and selecting features that consistently perform well despite differences in observer input. Variations arising from different scanners and acquisition protocols are more challenging to address and typically require repeated scans for thorough evaluation. Unfortunately, the limited availability of repeated scan data often restricts the ability to assess the reproducibility of radiomic features in a dataset-specific manner.

A review of the literature may provide insights into which features are reproducible. However, the transferability of feature repeatability across different studies is not always clear, and the absence of a standardized repeatability index for individual features can complicate the identification of robust features. Additionally, calibration of a radiomic model before its application in new clinical settings is strongly recommended to enhance its adaptability and performance.

In summary, the development of a generalizable radiomic model requires careful consideration of potential variations and the selection of features that are resistant to these changes. By incorporating robust preprocessing steps, involving multiple observers in segmentation, and calibrating the model for different settings, researchers can improve the reliability and applicability of radiomic signatures across various clinical environments.

### 4.6. Enhancing the Reporting of Repeatability and Reproducibility in Radiomic Feature Studies

While research into the repeatability and reproducibility of radiomic features has significantly enhanced our understanding of their sensitivity to random effects and biases, leveraging this knowledge to bolster the repeatability and reproducibility of radiomic signatures is paramount. In the developmental stages of radiomic signatures, the feasibility of conducting supplementary test–retest scans for dataset-specific assessments is often limited. Therefore, it becomes crucial for studies focusing on repeatability and reproducibility to explore whether their findings can aid other researchers in evaluating the repeatability and reproducibility of their own radiomic models, especially in scenarios where additional test–retest scans are not available. To support this endeavor, we propose the following specific recommendations:(1)**Detailed Reporting of Variation Sources**: Authors should meticulously document any sources of variation encountered across different measurement settings. These include, but are not limited to, changes in scanner types, imaging protocols, and segmentation processes. Such detailed reporting will provide valuable context for understanding the conditions under which the radiomic features were assessed.(2)**Transparent Disclosure of Calculation Parameters**: It is imperative to transparently disclose all parameters used in the calculation of radiomic features. This transparency ensures that other researchers can accurately replicate the feature extraction process, facilitating a more reliable comparison of results across different studies.(3)**Careful Selection of a Suitable Reliability Index**: Choosing an appropriate reliability index is critical for assessing the repeatability and reproducibility of radiomic features. Researchers should select indices that most accurately reflect the nature of the variations.(4)**Comprehensive Reporting of Reliability Metrics**: The reliability metrics for individual features should be thoroughly reported. This comprehensive reporting will allow other researchers to discern which features are most stable and reliable across different datasets and conditions, thereby informing the selection of features for their own radiomic signatures.

By adhering to these recommendations, researchers can facilitate a more precise evaluation of the repeatability and reproducibility of radiomic signatures, even in scenarios where direct test–retest data are unavailable. This approach not only advances the field of radiomics by ensuring the development of more robust and reliable signatures, but also fosters a culture of transparency and reproducibility within the research community.

Positron emission tomography (PET) is another crucial imaging modality in radiology. Unlike CT and MRI, which provide anatomical images with clear tissue boundaries, PET is a functional imaging technique that relies on the type of radiopharmaceutical tracer used. PET is also significant in radiomic research [59]. The concepts discussed in this review can be applied to identify highly repeatable PET features. 

## 5. Conclusions

In conclusion, the exploration of repeatability and reproducibility in radiomic features has significantly deepened our understanding of their susceptibility to both random effects and systematic biases. This knowledge is indispensable for the advancement of radiomic research, particularly in the development of robust and reliable radiomic signatures that can withstand the variability inherent in clinical settings. However, the practical challenges of conducting additional test–retest scans for dataset-specific evaluations highlight the necessity of a standardized approach in reporting and assessing the repeatability and reproducibility of radiomic features.

To address these challenges, we have proposed a set of recommendations aimed at enhancing the transparency and reliability of radiomic studies. These include the detailed reporting of sources of variation, transparent disclosure of feature calculation parameters, careful selection of reliability indices, and comprehensive reporting of reliability metrics for individual features. Adherence to these guidelines will not only facilitate more accurate evaluation of radiomic signatures in the absence of extensive test–retest data but also contribute to the broader goal of achieving generalizable and clinically applicable radiomic models.

## Figures and Tables

**Figure 1 diagnostics-14-01835-f001:**
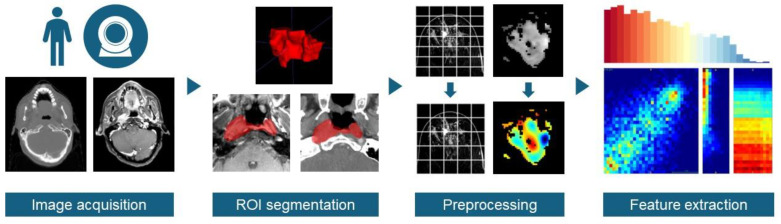
Steps of radiomic feature extraction that could affect radiomic feature values. The red color in ROI segmentation indicates the gross tumor volume as target in Radiotherapy. The color spectrum of Preprocessing indicates the high Hounsfield units (red) and low Hounsfield units (blue) within the gross tumor volume. The color spectrum of Feature extraction indicates the varied features value.

**Figure 2 diagnostics-14-01835-f002:**
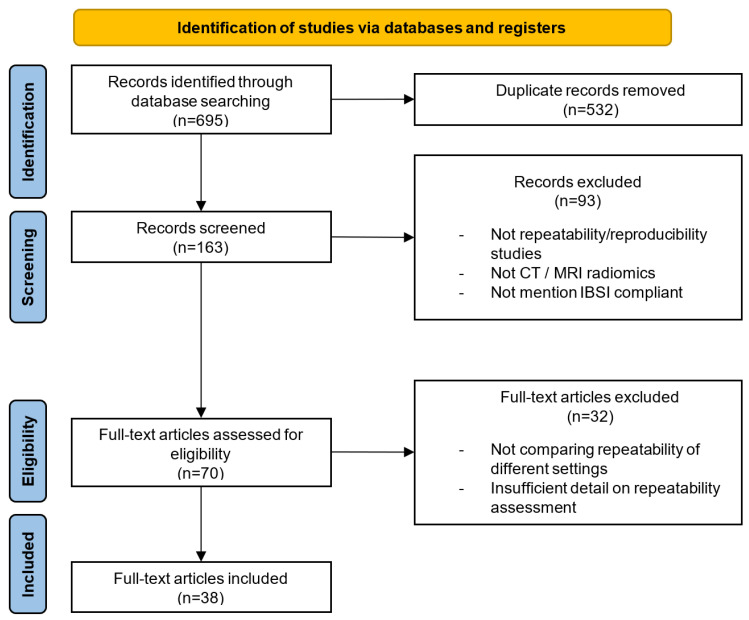
Flowchart of the selection of publications for the review.

**Table 1 diagnostics-14-01835-t001:** Summary of the literature on random effects affecting CT-based radiomic features.

Author	Modality	Sources of Variation	Criteria for High Repeatablity/Reproduciblity	Site	Highly Repeatable/Reproducible Features	Availability of Reliability Index
Chen et al. (2021) [27]	CT	Intra-scanner test–retest	CV < 10%	Phantom and hematoma	Phantom: 79.05% to 81.43% Hematoma: 42.54% to 45.4%	No
Chen et al. (2022) [22]	DECT SECT	Intra-scanner test–retest	Bland–Altman analysis > 0.90	Phantom	DECT: 87.02 ± 5.79% SECT: 92.91 ± 1.89%	Yes
Euler et al. (2021) [19]	CT	Intra-scanner test–retest	CCC and DR ≥ 0.9	Liver	74% to 86% repeatable features under acquisition settings	No
Mahon et al. (2019) [28]	CT	Intra-scanner test–retest	CCC > 0.9	Lung	Tumor: 54.4% Normal tissue: 78.5%	Yes
Muenzfeld et al. (2021) [29]	CT	Intra-scanner test–retest	CCC > 0.85	Phantom	19/86 (22%)	Yes
Prayer et al. (2020) [30]	CT	Intra-scanner test–retest	ICC > 0.9	Lung	65/86 (75.58%)	Yes
Duan et al. (2022) [31]	MIP	Intra-observer variability	ICC > 0.75	Liver	77/107 (71.96%)	Yes
	CECT	Intra-observer variability			84/107 (78.50%)	
Kocak et al. (2019) [32]	CECT	Intra-observer variability	ICC > 0.75	Kidney	Texture features: 693/744 (93.1%)	No
	CT	Intra-observer variability			Texture features: 686/744 (92.2%)	
EPV Le et al. (2021) [33]	CT	Perturbations	ICC > 0.9	Heart	14/93 (15.1%)	No
Müller-Franzes et al. (2022) [34]	CT	Autogenerated segmentations	ICC > 0.99	Multi-site	Lung: 269/439 (61.28%) Liver: 292/439 (66.51%) Kidney: 377/439 (85.88%)	Yes

Abbreviation: MIP, maximum intensity projection; CT, computed tomography; CECT, contrast-enhanced computed tomography; CV, coefficient of variation; CCC, concordance correlation coefficient; ICC, intra-class coefficient of correlation; DR, dynamic range.

**Table 2 diagnostics-14-01835-t002:** Summary of the literature on bias affecting CT-based radiomic features.

Author	Modality	Sources of Variation	Criteria for High Repeatablity/ Reproduciblity	Site	Highly Repeatable/Reproducible Features	Availability of Reliability Index
Chen et al. (2021) [27]	CT	Acquisition parameters	CV < 10%	Phantom and hematoma	Phantom: 48.89% to 53.97% Hematoma: 43% to 42.38%	No
Chen et al. (2022) [22]	CT	Acquisition parameters	ICC/CCC > 0.90	Phantom	DECT: 10.76 ± 2.05% SECT: 10.28 ± 2.05%	Yes
		Inter-scanner variability	CV/QCD < 10%		DECT: 15.16 ± 3.26%, 32.78 ± 5.62% SECT: 17.09 ± 2.60%, 27.73 ± 4.07%	
Denzler et al. (2021) [35]	CT	Acquisition parameters	ICC > 0.9	Lung	360/1386 (26%)	Yes
Euler et al. (2021) [19]	CT	Acquisition parameters	CCC and DR ≥ 0.9	Liver	32.7% to 99.2% reproducible features across different energies	No
Gruzdev et al. (2020) [36]	CECT	Inter-observer variability	Kendall’s concordance coefficient > 0.7	Pancreas	52/52 (100%) features for all phases	No
		Inter-scanner variability			74% reproducible texture features	
		Inter-scanner and inter-observer variability			67% reproducible texture features	
Ibrahim et al. (2021) [37]	CT	Contrast-enhanced phases	CCC > 0.9	Liver	42/167 (25.15%)	No
Lennartz et al. (2022) [38]	DECT	Inter-scanner variability	CCC > 0.9	Phantom and multi-sites	Phantom: None Patients: 2.5% to 16.1% of features	No
Meyer et al. (2019) [39])	CT	Acquisition parameters	R^2^ ≥ 0.95	Liver	12/106 (11%)	Yes
Muenzfeld et al. (2021) [29]	CT	Acquisition parameters	CCC > 0.85	Phantom	11/86 (12.8%)	Yes
Perrin et al. (2018) [40]	CECT	Contrast-agent injection rate	CCC > 0.9	Liver	Liver parenchyma: 63/254 (24.8%) and 0/254 (0%) Liver malignancies: 68/254 (26.77%) and 50/254 (19.69%)	Yes
		Acquisition parameters			Liver parenchyma: 20/254 (7.87%), 0/254 (0%); Liver malignancies: 34/254 (13.39%)	
Prayer et al. (2020) [30]	CT	Inter-scanner variability	ICC > 0.9	Lung	ICC ranges from 0.471 to 0.927	Yes
Refaee et al. (2022) [41]	CT	Acquisition parameters	CCC > 0.9	Phantom	6/91 (6.59%) to 78/91 (85.71%)	No
Rinaldi et al. (2022) [42]	CT	Acquisition parameters	OCCC ≥ 0.85	Lung	1260/1414 (89.11%)	Yes
Bianconi et al. (2021) [43]	CT	Inter-observer variability	Average symmetric mean absolute percentage error < 10%	Lung	30/88 (34.09%)	Yes
Duan et al. (2022) [31]	MIP	Inter-observer variability	ICC > 0.75	Liver	71/107 (66.36%)	Yes
	CECT	Inter-observer variability			74/107 (69.16%)	
Haarburger et al. (2020) [44]	CT	Inter-observer variability and automatic segmentation	ICC > 0.9	Multi-site	Lung: 71/105 (67.62%) Kidney: 61/105 (58.10%) Liver: 75/105 (71.43%)	Yes
Kocak et al. (2019) [32]	CECT	Inter-observer variability	ICC > 0.75	Kidney	Texture features: 632/744 (84.9%)	No
	CT	Inter-observer variability			Texture features: 571/744 (76.7%)	
Konik et al. (2021) [45]	CT	Inter-observer variability	ICC > 0.85	Kidney	78/169 (46.15%)	Yes
Li et al. (2020) [23]	CT	Preprocessing parameters	ICC > 0.8 and CV < 20%	Phantom	44/88 (50%)	No
Le et al. (2021) [33]	CT	Preprocessing parameters	ICC > 0.9	Heart	52/93 (55.9%)	No
Bianconi et al. (2021) [43]	CT	Preprocessing parameters	Averaging symmetric mean absolute percentage error < 10%	Lung	28/88 (31.82%)	Yes

Abbreviation: MIP, maximum intensity projection; CT, computed tomography; CECT, contrast-enhanced computed tomography; CV, coefficient of variation; CCC, concordance correlation coefficient; ICC, intra-class coefficient of correlation; OCCC, overall concordance correlation coefficient.

**Table 3 diagnostics-14-01835-t003:** Summary of the literature investigating random effects affecting MR-based radiomic features.

Author	Modality	Sources of Variation	Criteria for High Repeatablity/Reproduciblity	Site	Highly Repeatable/Reproducible Features	Availability of Reliability Index
Carbonell et al. (2022) [20]	MRI	Intra-scanner test–retest	ICC > 0.9 and CV < 20%	Liver	HCC-T1WIpre: 45/108 (41.67%), T1WIpvp: 47/108 (43.52%), T2WI: 39/108 (36.11%), ADC: 21/108 (19.44%) Liver-T1WIpre: 32/92 (34.78%), T1WIpvp: 16/92 (17.39%), T2WI: 12/92 (13.04%), ADC: 2/92 (2.17%)	Yes
Fiset et al. (2019) [13]	MRI (T2WI)	Intra-scanner test–retest	ICC ≥ 0.75	Cervical	Cervical: 917/1761 (52.1%)	Yes
Mahon et al. (2019) [28]	MRI	Intra-scanner test–retest	CCC ≥ 0.9	Lung	Lung (TRUFISP): 64.4% (tumor), 67.8% (normal tissue) Lung (VIBE): 54.4% (tumor), 72.9% (normal tissue)	Yes
Mitchell-Hay et al. (2022) [21]	MRI (T1WI)	Intra-scanner test–retest	CCC/DR > 0.9	Brain	8/1596 (0.50%) features were repeatable in all centers	Yes
Pandey et al. (2021) [46]	MRI (T2WI)	Intra-scanner test–retest	ICC > 0.5	Brain	ICC: 0.73 for right grey matter, 0.78 for left grey matter ICC: 0.65 for right white matter, 0.67 for left white matter	Yes
Dewi et al. (2023) [47]	MRI (T2WI)	Intra-scanner test–retest	ICC > 0.75	Prostate	25/107 (23.36%) at fixed bin count discretization of 64	Yes
Duan et al. (2022) [31]	MRI	Intra-observer variability	ICC > 0.75	Liver	98/107 (91.6%)	Yes
Müller-Franzes et al. (2022) [34]	MRI	Automatic segmentations	ICC > 0.99	Brain	77/439 (17.54%)	Yes

Abbreviations: CCC, concordance correlation coefficient; ICC, intra-class coefficient of correlation; CV, coefficient of variation; T1WI, T1-weighted image; T2WI, T2-weighted image; ADC, apparent diffusion coefficient; HCC, hepatocellular carcinoma.

**Table 4 diagnostics-14-01835-t004:** Summary of the literature investigating bias affecting MR-based radiomic features.

Author	Modality	Sources of Variation	Criteria for High Repeatablity/Reproduciblity	Site	Highly Repeatable/Reproducible Features	Availability of Reliability Index
Carbonell et al. (2022) [20]	MRI	Inter-observer variability	CCC > 0.75 and CV < 20%	Liver	HCC-T1WIpre: 95/108 (87.96%), T1WIpvp: 102/108 (94.44%), T2WI: 61/108 (56.48%), ADC: 91/108 (84.26%) Liver-T1WIpre: 25/92 (27.17%), T1WIpvp: 37/92 (40.22%), T2WI: 8/92 (8.70%), ADC: 49/92 (53.26%)	Yes
		Inter-scanner variability	CCC > 0.75, CV < 20%		HCC-T1WIpre: 23/108 (21.30%), T1WIpvp: 25/108 (23.15%), T2WI: 11/108 (10.19%), ADC: 7/108 (6.48%) Liver-T1WIpre: 0/92 (0%), T1WIpvp: 0/92 (0%), T2WI: 0/92 (0%), ADC: 0/92 (0%)	
Fiset et al. (2019) [13]	MRI (T2W)	Inter-observer variability	ICC > 0.9	Cervix	1301/1761 (73.88%)	Yes
		Inter-scanner variability	ICC ≥ 0.75		248/1761 (14.1%)	Yes
Lee et al. (2021) [24]	MRI	Acquisition parameters	ICC > 0.9, CV < 20%	Phantom and brain (healthy volunteers)	Phantom: average ICC, 0.963 (T1WI) and 0.959 (T2WI) Brain: average ICC, 0.856 (T1WI) and 0.859 (T2WI)	Yes
Mitchell-Hay et al. (2022) [21]	MRI (T1WI)	Inter-scanner variability	ICC > 0.9	Brain	40/1595 (2.51%) features were excellent in terms of reproducibility	Yes
Pandey et al. (2020) [46]	MRI (T2WI)	Spatial variability	ICC > 0.5	Brain	29.04% of gray matter and 38.7% of white matter features demonstrated an ICC > 0.5	Yes
		Inter-scanner variability			18% of gray matter and 21.5% of white matter features demonstrated an ICC > 0.5	
Raisi-Estabragh et al. (2020) [48]	MRI	Inter-scanner variability	ICC > 0.9	Cardiac	LV myocardium: 4/16 (25%) for repeatable shape features, (28/38, 74%) for repeatable first order features, (125/146, 86%) for repeatable texture features	Yes
Duron et al. (2019) [49]	MRI	Preprocessing parameters	ICC > 0.8 and CCC > 0.9	Lacrimal gland	54/69 (78.26%) for T2WI, 37/69 (53.62%) for T1WI, and 31/69 (44.93%) for ADC	No
				Breast	32/69 (46.38%) for DISCO sequence	
Moradmand (2019) [26]	MRI	Preprocessing parameters	CCC/DR > 0.9	Brain (glioblastoma)	703/4066 (17.3%)	No
Scalco et al. (2020) [50]	T2w-MRI	Preprocessing parameters	ICC > 0.9	Prostate	Prostate: 14% Obturators: 12% Bulb: 13/91 (14%)	Yes
Duan et al. (2022) [31]	MRI	Inter-observer variability	ICC > 0.75	Liver	85/107 (79.4%)	Yes
Fiset et al. (2019) [13]	MRI (T2W)	Inter-observer variability	ICC > 0.9	Cervix	1301/1761 (73.88%)	Yes
Haniff (2021) [51]	MRI	Semi-automatic segmentation	ICC ≥ 0.8	Liver	640/662 (96.7%)	Yes (partial)
		Inter-observer variability			517/662 (78.1%)	
Müller-Franzes et al. (2022) [34]	MRI	Automatic segmentations	ICC > 0.99	Brain	77/439 (17.54%)	Yes

Abbreviations: CCC, concordance correlation coefficient; ICC, intra-class coefficient of correlation; CV, coefficient of variation; T1WI, T1-weighted image; T2WI, T2-weighted image; ADC, apparent diffusion coefficient; HCC, hepatocellular carcinoma.

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
