# Peer review of "Enhancing the Clinical Utility of Radiomics: Addressing the Challenges of Repeatability and Reproducibility in CT and MRI"

_diagnostics, 2024, doi:10.3390/diagnostics14161835_

Round 1

Reviewer 1 Report

Comments and Suggestions for Authors

In their review 'Enhancing the Clinical Utility of Radiomics: Addressing the
Challenges of Repeatability and Reproducibility', the authors review papers commenting on the repeatability and reproducibility of radiomic features. However, while radiomics is used for MR, CT, and PET data, no PET work occurs in the manuscript. While for PET feature reproducibility is especial important.

Additionally, I can also not see a real novelty as already two reviews about radiomics repeatability/Reproducibility exists that cover very similar aspects in their work. (Traverso et al., Pfaehler et al.)

Author Response

In their review 'Enhancing the Clinical Utility of Radiomics: Addressing the
Challenges of Repeatability and Reproducibility', the authors review papers commenting on the repeatability and reproducibility of radiomic features. However, while radiomics is used for MR, CT, and PET data, no PET work occurs in the manuscript. While for PET feature reproducibility is especial important.

Additionally, I can also not see a real novelty as already two reviews about radiomics repeatability/Reproducibility exists that cover very similar aspects in their work. (Traverso et al., Pfaehler et al.)

Reply: Thank you for your comments.

We agree that PET is a significant and widely studied imaging modality in radiomics research. While CT and MR provide anatomical images with clear tissue boundaries, PET is a functional imaging technique that depends heavily on the type of nuclear tracer used. Therefore, this review focuses on the repeatability and reproducibility of CT and MR radiomics features, providing detailed summaries of the sources of variation and the number of repeatable/reproducible features. We believe a separate review article is more appropriate for discussing PET radiomics due to its unique characteristics. Including PET results would significantly increase the length and complexity of this review, making it harder for the general audience to follow. To clarify the scope and acknowledge the importance of PET imaging in radiomics, we have modified the manuscript title and added a paragraph in the Discussion section addressing PET-related work, and cited the articles you provided. Line 384 to 387.

Regarding the novelty of this review, we acknowledge that the topic of radiomics feature repeatability and reproducibility has been extensively discussed in previous systematic reviews by Traverso et al. (2018) and Pfaehler et al. (2021). However, significant new research has emerged in the past three years, warranting an updated review to highlight recent findings and compare them with existing evidence. Our review provides more structured and detailed information, such as sources of variation, criteria for identifying reliable radiomics features, and the number/portion of identified reliable features, rather than general qualitative summaries. This detailed information is crucial for understanding the different impacts of random and bias effects on feature repeatability/reproducibility and for inspiring new approaches to address these issues in clinical applications. Additionally, we offer an in-depth discussion with actionable insights on how to practically evaluate and utilize radiomics feature repeatability/reproducibility in clinical research.

Reviewer 2 Report

Comments and Suggestions for Authors

This is a really useful study, which address the paucity of discussion regarding the factors that influence the reproducibility and repeatability of radiomic features. However, the reviewer has the following concerns.

(1) What type of database is used for data collection, which determines the accuracy of the study.

(2) In line 144~145, it was for MRI scans?

(3) In line 172, is it CT-based or MRI-based radiomics features?

(4) In 4.1 part, it should focus on the significant instead of the study aim.

(5) Please avoid using first-person point of view as much as possible when writing.

(6) It would be better if there is one CT or MRI figure.

Comments on the Quality of English Language

The English writing is good, and the ideas are presented in a logical order, adhering to the rules of English grammar, although there are some minor errors.

Author Response

This is a really useful study, which address the paucity of discussion regarding the factors that influence the reproducibility and repeatability of radiomic features. However, the reviewer has the following concerns.

Thank you very much for the comments, and we appreciate the positive feedback from you.

(1) What type of database is used for data collection, which determines the accuracy of the study.

Thank you for the question. Three databases were used to gather the records of the publication, PubMed, Embase and Web of Science. Then, we employed criteria to select the studies: 1) peer-reviewed English full-text reports; 2) having radiomics features extracted from CT images or MR images; 3) indicating compliance with IBSI during feature extraction; and (4) focusing on the repeatability and reproducibility of radiomic features resulting from variations during image acquisition and segmentation.

To clarify the issue, we visualized our selection workflow, as Figure 2 in the main manuscript.

(2) In line 144~145, it was for MRI scans?

Reply: Thank you very much for citing such error, yes, we missed “for MRI scans”. Thank you very much, and we have added it in line 144~145.

(3) In line 172, is it CT-based or MRI-based radiomics features?

Reply: Thank you for spotting the error. It should be MRI-based radiomics features.

(4) In 4.1 part, it should focus on the significant instead of the study aim.

Reply: Thank you for the constructive comments. We agree that the first paragraph of discussion should focus on the significance. Therefore, we modified the first paragraph of discussion (part 4.1) and emphasized the significance of this review.

 “Radiomics has emerged as a pivotal technique to augment the value of medical imaging through a high-throughput characterization of medical images. The explicit mathematical definitions of each radiomic feature enhance the interpretability of the radiomic signature, offering a more transparent alternative to the "black box" deep learning models. An increasing number of publications have explored the diagnostic and prognostic value of radiomic approaches for various diseases in recent years. However, despite the surge in publications, concerns about reproducibility and repeatability have been prevalent since the inception of the field. It is believed that the usage of high repeatable and reproducible radiomics features should be the first and fore-most criteria to safeguard downstream model reliability. Previous evidence has also suggested the positive impact of repeatable radiomics features in improving both internal and external generalizability of radiomics models. Guidelines such as the EvaluAtion of Radiomics research (CLEAR) [52], Radiomics Quality Score (RQS) [53], and the Joint EANM/SNMMI guideline on radiomics [54] have underscored the importance of evaluating the reproducibility and repeatability of radiomic features and models. Understanding the source of radiomics feature variability is fundamental to harnessing the potential of radiomics in precision medicine by ensuring its reliability and determining its scope of application.

(5) Please avoid using first-person point of view as much as possible when writing.

Reply: Thank you for pointing this out. We scanned the entire manuscript, and modified the paragraph with first person point of view, which include: line 100, line 103, line 106, and line 348.

(6) It would be better if there is one CT or MRI figure.

Reply: Thank you very much for the suggestion. We have added a general workflow of radiomics feature extraction that contains demonstrations of CT and MR images in the revised manuscript.

Reviewer 3 Report

Comments and Suggestions for Authors

Review of “Enhancing the Clinical Utility of Radiomics: Addressing the Challenges of Repeatability and Reproducibility”

The review aims to identify the criteria for repeatability and reproducibility of radiomic features extracted from two principal types of imaging, specifically CT and MR, on domains of different pathologies. In particular, the review focuses on the systematic analysis of the main types of errors, bias and random, that can happen during the various steps of extracting and calculating radiomic features from the image. For this, the researchers found a satisfying number of scientific papers that they compared by analyzing the main metrics for feature reproducibility and repeatability.

The review is clear, understandable, and above all very detailed especially in the way the results were reported and how they were discussed. There are some minor corrections that could be made to clarify some concepts and facilitate more intuitive reading. They are divided per sections, as follows:

Introduction - A good overview is given of the rationale, background, and current state of the art in the use of radiomics. Indeed, the introduction is clearly written and well argued.

In line 42, the authors could add more references that may be of interest to the reader to get an overview of the application fields (e.g., breaking down the use of radiomics to identify different diseases such as Alzheimer’s (Radiomics and Artificial Intelligence for the Diagnosis and Monitoring of Alzheimer's Disease: A Systematic Review of Studies in the Field – DOI: 10.3390/jcm12165432 ) or Parkinson’s (Improved prediction of outcome in Parkinson's disease using radiomics analysis of longitudinal DAT SPECT images - DOI: 10.1016/j.nicl.2017.08.021).  

Materials and methods - Again, the methodology is simple and well defined. It would be helpful to add only a flowchart indicating the main steps of the research. It might promote more immediate and intuitive reading. A template is left to follow:

Results - It is a well-described and especially organically managed section divided into chapters.

In Tables 1 and following, the authors should find a way to make some columns in the table more readable, especially the column "Results for high repeatable/reproducible features." The authors should find a way to standardize the reported values since they are different in nature and difficult to find a common pattern among all of interpretation.

Discussion – at line 190, authors say that DL works as a black box. It would be interesting if the authors included a comment on the main differences between an automatic DL feature extraction (from the image) and a manual radiomic extraction with ML.

In line 239 when the authors say "inherent biases," it should be explained further. What is meant by different scanners or observer practice? Are there also other factors to be taken into consideration?

At line 269 the authors say "First, the extraction process should be refined to maximize the number of reproducible features, which involves optimizing image preprocessing and segmentation techniques."  What types of preprocessing and segmentation? Explain further.

Author Response

The review aims to identify the criteria for repeatability and reproducibility of radiomic features extracted from two principal types of imaging, specifically CT and MR, on domains of different pathologies. In particular, the review focuses on the systematic analysis of the main types of errors, bias and random, that can happen during the various steps of extracting and calculating radiomic features from the image. For this, the researchers found a satisfying number of scientific papers that they compared by analyzing the main metrics for feature reproducibility and repeatability.

The review is clear, understandable, and above all very detailed especially in the way the results were reported and how they were discussed. There are some minor corrections that could be made to clarify some concepts and facilitate more intuitive reading. They are divided per sections, as follows:

Introduction - A good overview is given of the rationale, background, and current state of the art in the use of radiomics. Indeed, the introduction is clearly written and well argued.

In line 42, the authors could add more references that may be of interest to the reader to get an overview of the application fields (e.g., breaking down the use of radiomics to identify different diseases such as Alzheimer’s (Radiomics and Artificial Intelligence for the Diagnosis and Monitoring of Alzheimer's Disease: A Systematic Review of Studies in the Field – DOI: 10.3390/jcm12165432 ) or Parkinson’s (Improved prediction of outcome in Parkinson's disease using radiomics analysis of longitudinal DAT SPECT images - DOI: 10.1016/j.nicl.2017.08.021).  

Reply: Thank you for your comments, the references that you provided especially for Alzheimer’s disease and Parkinson’s disease would broaden the audience and widen the application scenario of this review. Therefore, we gladly added the references provided by the reviewer. Therefore, we added citations in introduction line 35, citation number 3-4.

Materials and methods - Again, the methodology is simple and well defined. It would be helpful to add only a flowchart indicating the main steps of the research. It might promote more immediate and intuitive reading. A template is left to follow:

Results - It is a well-described and especially organically managed section divided into chapters.

In Tables 1 and following, the authors should find a way to make some columns in the table more readable, especially the column "Results for high repeatable/reproducible features." The authors should find a way to standardize the reported values since they are different in nature and difficult to find a common pattern among all of interpretation.

 --

Reply: Thank you very much for pointing this out. During the writing of the manuscript, we were struggled to have a term to describe this column. We would take your comments valuable and re-name the columns. The new columns name were 1) author, 2) modality, 3) sources of variation, 4) criteria for repeatability/reproducibility, 5) High repeatable/reproducible features, 6) Availability of reliability index. After the revision, we hope it would be more readable for the audience. Thank you for pointing out the lengthy of the table.

Discussion – at line 190, authors say that DL works as a black box. It would be interesting if the authors included a comment on the main differences between an automatic DL feature extraction (from the image) and a manual radiomic extraction with ML.

In line 239 when the authors say "inherent biases," it should be explained further. What is meant by different scanners or observer practice? Are there also other factors to be taken into consideration?

Reply: Thank you for your questions. They will substantially enhance the quality of the review. Several questions were raised, and we will address them one by one.

The first question concerns the comparison between deep learning (DL) and radiomics with machine learning. The primary distinction between DL feature extraction and radiomics feature extraction is that DL features depend on the network used for extraction, while radiomics features are calculated using explicit mathematical formulations. Currently, the advantage of manual calculation in radiomics is the standardization of features, thanks to Zwanenburg et al.’s Image Biomarker Standardization Initiative. This standardization ensures consistent measurement of radiomics features with identical images and segmentations. In contrast, the reproducibility of DL features is not guaranteed due to the lack of standardization. This raises a fundamental concern: can consistent DL feature values be generated across different research teams given the same set of images and segmentations and same deep learning network? The term "black box" refers to the challenge of standardizing DL procedures. This discussion has been added to the main manuscript's discussion section.

The second question pertains to inherent biases. This description emphasizes that the measurement of radiomics features is inherently biased. For example, if a radiomics score/model is constructed using features calculated from segmentations delineated by Radiologist A, the model will be biased towards Radiologist A. Such a model would achieve optimal performance only when segmentations are delineated by Radiologist A. The inherent bias is that the radiomics model would underperform in the absence of Radiologist A. Another way to understand the bias was to understand the differences between development stage and application stage. Therefore, we emphasize the importance of recognizing and understanding potential biases during radiomics feature measurement in this review. Other factors contributing to bias include scanner brand, acquisition parameters, and preprocessing parameters.

At line 269 the authors say "First, the extraction process should be refined to maximize the number of reproducible features, which involves optimizing image preprocessing and segmentation techniques."  What types of preprocessing and segmentation? Explain further.

Reply: Thank you for the question; it is crucial to clarify this point. Firstly, publications have suggested that preprocessing sequences impact the reproducibility of MRI-based features [Moradmand, H., Aghamiri, S.M.R., and Ghaderi, R. (2020), Impact of image preprocessing methods on reproducibility of radiomic features in multimodal magnetic resonance imaging in glioblastoma. J Appl Clin Med Phys, 21: 179-190. https://doi.org/10.1002/acm2.12795]. They conducted a comprehensive analysis comparing radiomics feature values from baseline images and preprocessed images for glioblastoma. This study raises the question of whether optimizing preprocessing sequences/parameters—such as interpolation, post-acquisition processing, rounding intensities, and re-segmentation/outlier filtering—can improve the number of highly repeatable features against random effects. This process is analogous to hyper-parameter tuning in deep learning, aimed at maximizing the number of repeatable features.

Regarding segmentations, manual and automatic segmentation are the primary methods. Given the challenge of replicating segmentation methods in practical applications or inter-institutional scenario, it is inappropriate to claim that optimizing segmentation techniques can maximize the number of radiomics features. We have corrected this description in the manuscript.

We thank you again for raising this question.